# Nudging individuals' creativity using social labeling

**Marine Agogué** [1]* , **Béatrice Parguel** [2]

**1** HEC Montréal, 3000 chemin Cote Ste Catherine, Montréal, QC, Canada, **2** Université Paris-Dauphine, PSL Research University, CNRS, DRM, Place du Maréchal de Lattre de Tassigny, Paris, France

◉ These authors contributed equally to this work.
* marine.agogue@hec.ca

**Data Availability Statement:** All relevant data are within the manuscript and its Supporting Information files.

**Funding:** This work was supported by the Idex Paris Science Lettres' ECMI grant (ANR-10-IDEX-0001-02 PSL*) and by the FRQSC (Grant #

## Abstract

Simple instructions have been shown to robustly influence individual creativity, which is key to solve local problems. Building on social labeling theory, we examine the possibility of nudging individual's creativity using "creative" and "not creative" labels. Study 1 showed that subjects labeled as "creative" or "not creative" performed better in a creative task than unlabeled subjects and established the moderating effect of self-perceived creativity. Among subjects scoring low on self-perceived creativity, those labeled as "creative" performed better than those labeled as "not creative". Conversely, among subjects scoring high on self-perceived creativity, those labeled as "not creative" tend to perform better than those labeled as "creative". Study 2 and Study 3 further explored the psychological mechanisms at play in both cases: specifically, Study 2 showed that applying a "creative" label has the ability to increase creative self-efficacy through self-perceived creativity, whereas Study 3 demonstrated that applying a "not creative" label has the ability to increase individual creativity performance through a higher involvement in the creative task.

## Introduction

Potential antecedents of collaborators' creativity, i.e. their ability to produce ideas that are both original and useful [1] [2], include collaborator's education level, learning orientation and job self-efficacy, supervisor's support and expectations, job complexity and creativity requirements, a favorable organizational climate or shared knowledge of who knows what [3] [4] [5] [6] [7]. Still, these drivers of individual creativity require such deep organizational changes that exploring how managers can activate individual creativity in a more direct and frugal way is a central issue. In this perspective, understanding whether self-perception based techniques, such as social labeling [8], can indirectly sustain collaborators' individual creativity contributes to our knowledge of essential psychological processes within organizations.

Social labeling is "*a persuasion technique that consists in providing a person with a statement about his or her personality or values (i.e. social label) in an attempt to provoke behavior that is consistent with the label*" [9]. Bem's re-attribution mechanism [10] explains the persuasion at stake in social labeling. Although self-observation of an individual's own behavior provides the

205466). The funders had no role in study design, data collection and analysis, decision to publish, or preparation of the manuscript.

**Competing interests:** The authors have declared that no competing interests exist.

clearest information to make self-attribution of dispositional properties [11], labels provided by others can also be informative about a person's traits and values and lead to their re-attribution to the self [12]. Scholars have theorized that a labeled individual appears to internalize the values or personality traits associated with the label as representative of his or her basic self-perception, leading to changes in subsequent behaviors [13] [14] [15]. To date, however, no research has specifically interrogated whether social labeling may play a role in enhancing creative self-efficacy perceptions, hence boosting individual creativity. The question of whether labeling an individual as "creative" (or "not creative") supports individual creativity remains open.

To explore the impact of labeling a collaborator as "creative" (or "not creative") on individual creativity performance, we conducted three experiments that all began with a self-assessment creativity questionnaire to provide a credible basis for random labeling. A first study shows that participants labeled as "creative" or "not creative" people performed better in the creative task than those in the control condition. Going further, our results suggest a moderating effect of self-perceived creativity. Further interrogating these initial results, two follow-up studies investigated the mechanism at stake when assigning individuals to a "creative" or "not creative" label (versus no label). Study 2 shows that labeling individuals as "creative" enhances successively their self-perceived creativity and their creative self-efficacy, explaining why subjects scoring low on self-perceived creativity would perform better when labeled as "creative". Study 3 suggests that labeling individuals as "not creative" activates psychological reactance and enhances subjects' involvement in the creative task, hence individual creativity performance. These findings thus contribute to the literature on creativity management by offering relevant insights on new ways to nudge collaborators' creativity.

## Boosting individual creativity within organizations

Ideation is usually a demanding activity that requires time and cognitive efforts, while remaining a highly uncertain activity with a risk of failure. It is therefore paramount that individuals should have the confidence to remain persistent in creative ideation, thus developing creative self-efficacy, defined as the belief that one has the knowledge and skills to produce creative outcomes [6]. And it is the managers' job to shape the organization's culture [16] likely to disseminate key assumptions, beliefs, and values regarding creativity, thus providing psychological safety [17] and impacting individual ideation. Direct extrinsic stimuli from managers to employees–such as managerial instructions to "be creative"–have been shown to have a concrete impact on performance in tests of creative thinking [18] [19] [20]. Specifically, they may lead to more creative ideas, but to a lower number of ideas [21], although sometimes generating psychological reactance [22]. Those results are not always replicated and remain questionable [23], calling to further research on the issue of indirect stimuli provided by managers to foster individual creativity.

**Direct effect of the "creative" label.** Social labeling techniques have proven effective to elicit a variety of desirable behaviors (e.g. eco-friendly consumerism, scoring better in math, signing a petition, giving to charity) from a collection of targets such as children, students or adults (e.g. [24] [25]). As such, they could play as relevant indirect stimuli provided by managers to enhance individual creativity performance.

In line with the literature, we contend that labeling an individual as "creative" could alter his or her self-perception, leading to a view of the self as creative, as soon as the label appears plausible. To build a plausible label, experimenters usually rely on people's previous statements [12] [26] [27] or recent behavioral evidence [9] [28]. When the label does not appear as plausible, persuasion knowledge is activated and the label is rejected [29] [9]. As a conclusion, the

persuasive intent behind the label must be masked for the self-perception to be altered among adults. Such a modification of self-perception is then likely to boost his or her creative self-efficacy. Framed in terms of psychological empowerment, general self-efficacy is found to influence the motivation and ability to engage in, and cope with, specific tasks [30]. Similarly, creative self-efficacy enhances the motivation and ability to engage in, and cope with, creative tasks, resulting in enhanced individual creativity performance, as proposed (e.g. [30] [31]) and consistently confirmed (e.g. [3] [5] [6] [7] [32]) by the literature.

Based on the above, we argue that the application of a "creative" label will be effective if the subjects do not perceive the persuasion attempt behind the label. We therefore hypothesize the following:

**Hypothesis 1:** Subjects labeled as "creative" display higher creativity performance than unlabeled subjects, i.e. a control group.

**Direct effect of the "not creative" label.**   Social labeling effectiveness relies on the re-attribution to the self of the qualities stressed in the label, whether the label relates to positive or negative qualities [13] [8]. While re-attribution to the self of positive qualities consistently enhances positive behaviors (e.g. [9] [28]), re-attribution to the self of negative qualities has a less consistent influence on subsequent behaviors [12]. On the one hand, subjects could re-attribute a negative label to their negative qualities, and behave accordingly. Goldman and colleagues [14] showed that subjects labeled as unhelpful were less likely to comply with a request, suggesting that the negative label could have activated the perception of actually being a rather unhelpful person. This negative influence of negative labels is in line with the interactionists' perspective on deviance, whose central contribution to labeling theory is that treating an individual as if he or she were deviant leads that person to share this perception of him/herself and preserves his or her deviant behavior (e.g. [33]). On the other hand, subjects could re-attribute a negative label to their negative qualities, and make efforts to disprove it. To explain this, research suggests that negative labels could make the individual more sensitive to the consequences of being associated with negative qualities, the way he or she presents himself, and the way he or she is perceived, hence his or her self-image [8]. Steele [34], for example, showed that subjects labeled as individualistic are more likely to give some time to a charity in order to restore their self-esteem.

Negative labels can thus lead to two contradictory outcomes: confirmation of the negative label vs. resistance to the negative label. The acceptance or rejection of the label may depend on the ability of the subsequent requested behavior to effectively contradict the label [35]. Further testing this explanation by examining congruence between the label and the requested behavior, Guéguen [25] confirmed that subjects are more likely to reject a negative label when the label is linked to the nature of the subsequent requested behavior, thus restoring their self-esteem. In contrast, when the requested behavior has no link with the basis for the negative label, subjects have no opportunity to behave in a way that directly contradicts the negative label, thus behave in a way that confirms the label. Following, asking individuals to participate in a creative task clearly appears as a relevant opportunity to contradict a "not creative" label and demonstrate creativity.

Hence, a "not creative" label is expected to erode collaborators' self-esteem in terms of creativity, but should actually encourage them to invest efforts to display high creativity performance in a subsequent creative task in order to restore it. We therefore propose the following hypothesis:

**Hypothesis 2:** Subjects labeled as "not creative" display higher creativity performance than unlabeled subjects, i.e. a control group.

Hypotheses 1 and 2 propose that labeling a collaborator as "creative" or "not creative" will both result in an increase in individual creativity performance. These hypotheses are not mirror images of each other, being based on different psychological mechanisms, and are both expected to be supported.

**Moderating effect of self-perceived creativity.** The effectiveness of social labeling depends on the actual existence of a sufficient and clear self-perceptions in labeled subjects [24] [36] and on the label's ability to modify self-perceptions [13] [14] [15]. Social labeling effectiveness should therefore depend on the way individuals initially perceive themselves as regards the targeted quality. Actually, a "positive" label should provide more room for modification among people scoring low on the targeted quality than higher scorers, and thus have more effect on the former than the latter group. A "negative" label, in contrast, should provide less room for modification among people scoring low on the targeted quality than higher scorers, and thus have more effect on the latter than the former group. Following this rationale, "creative" and "not creative" labels are likely to influence individual creativity performance differently depending on their ability to modify the way subjects initially perceive themselves in terms of personal creativity. A "creative" label is particularly likely to enhance self-perception, creative self-efficacy, and individual creativity performance, among subjects initially perceiving themselves as lowly creative. In contrast, a "not creative" label is particularly likely to erode self-perception, motivate self-esteem restoration and enhance individual creativity performance, among subjects initially perceiving themselves as highly creative.

We therefore expect that a "creative" label should be more effective among subjects scoring low on self-perceived creativity, and a "not creative" label among subjects scoring high on self-perceived creativity. Our final hypothesis is thus that self-perceived creativity moderates the influence of the positive or negative nature ("creative" vs "not creative") of the label on individual creativity performance.

**Hypothesis 3:** The effectiveness of a "creative" vs. "not creative" label on subjects' creativity is moderated by subjects' self-perceived creativity, such that:

a. a "creative" label generates higher creativity performance among subjects scoring low on self-perceived creativity,

b. whereas a "not creative" label generates higher creativity performance among subjects scoring high on self-perceived creativity.

Fig 1 synthesizes the hypotheses of the conceptual model.

We will test our conceptual model in Study 1, before further exploring the psychological mechanisms at play when labeling an individual as "creative" in Study 2, i.e. the activation of creative self-efficacy, or when labeling an individual as "not creative" in Study 3, i.e. an enhanced involvement in the creative task. We will therefore test in two follow-up studies why both "creative" and "not creative" labels may enhance individual creativity performance.

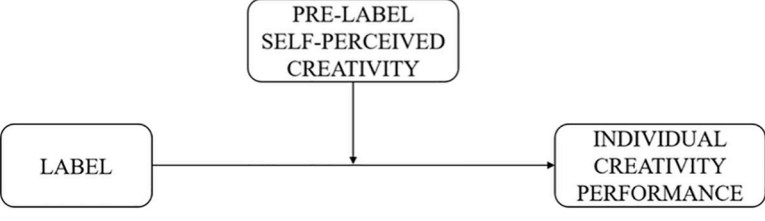

**Fig 1. Conceptual model.**

## Materials and methods

### Study 1

**Procedure.** The HEC Montreal's Ethic Committee approved this study (approval # 2015–62). Consent was obtained by the participant going through the questionnaire—any participant willing to stop could just opt out of the online questionnaire. Besides data were analyzed anonymously.

Before engaging in the study *per se*, participants registered their age, gender and education level, as these variables are traditionally considered as potentially influencing individual creativity performance (e.g. [3] [37] [6]).

Then, to ensure effective labeling of participants as "creative" or "not creative", we had to make our label credible, and so the participants were first asked to complete an initial questionnaire about their self-assessed ability to generate creative ideas. Participants were then randomly assigned to one of the four conditions: a control condition, a "creative" label, a "not creative" label, and a "moderately creative" label. The "moderately creative" label was introduced to rule out the potential effect on individual creativity performance of the mere application of a "creative" or "not creative" label, that could increase the salience or perceived importance of creativity behavior, and drive a person to act accordingly independently of the positive or negative nature of the label. Following, we expect that participants labeled as "moderately creative" will not have a higher creativity performance than unlabeled participants, i.e. a control group.

Concretely, three quarters of participants were shown their supposed positions on a graph in which: (1) the highest-scoring 10% of the population were given the "creative" label, (2) the lowest-scoring 10% were given the "not creative" label, and (3) the rest of the population were given the "moderately creative" label. Members of the three label conditions were told: "*Here is a histogram of the distribution of individual creativity scores of a representative sample of the population, allowing you to visualize your own level of creativity. The answers you just gave place you in the shaded column.*" In the control condition, participants were simply thanked for their answers and given no feedback regarding their responses to the creativity questionnaire. Fig 2 illustrates the experimental stimuli.

Of note, we acknowledge that disguising what we were studying deceives subjects but appears necessary to preserve the internal validity of our results. Though it can be considered as a big issue by economists, most social sciences that have been carrying experiments for decades tend to consider it as a legitimate procedure [38].

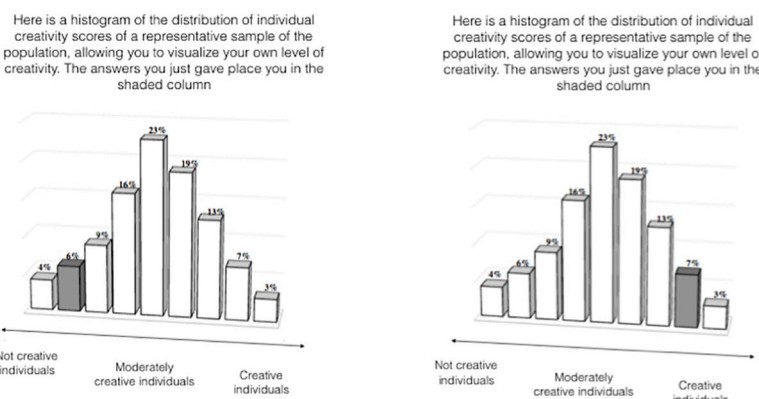

**Fig 2. Experimental stimuli.** "Not creative" label on the left. "Creative" label on the right.

After filling in the questionnaire and being labeled (or not), participants were asked to suggest a number of creative ideas for completing the "egg task". This divergent thinking task is commonly used in team-building warm-ups by practitioners [39] or in teaching (e.g. [40]), and has also been used in recent years in experimental studies of creativity (e.g. [41] [42] [43]). The egg task is presented as follows: "*Please suggest as many original solutions as possible to the following problem*: *Make sure that a hen's egg that is dropped from a height of 10 meters does not break*." All participants were given as long as they wanted to write down as many original solutions as they could, with a minimal answering time of 10 minutes. To conclude, we asked the participants who were labeled to recall the nature of the label they were provided with to check the validity of our manipulation. Disregarding the outliers (i.e. the 10% fastest and the 10% slowest respondents), completing the whole survey took between 12 and 22 minutes (mean = 15, s.d. = 2).

For ethical purposes, after completing the whole procedure, participants were informed in the following words that the label was purely randomly applied: "*Your creativity score stated at the start of the survey was randomly assigned for the purpose of our research. It does not reflect your actual level of creativity at all.*" As for Study 2 and Study 3, Study 1 experimental procedure was approved by one of the authors' Institutional Review Board.

**Measures.** Before applying the label, we measured *pre-label self-perceived creativity* in line with existing literature (e.g. [44] [45] [46]), using 6-items ranging from 1 "strongly disagree" to 7 "strongly agree" (see Table 1). This measure displays a good reliability (Cronbach's α = .849).

*Individual creativity performance* was measured in terms of originality, i.e. the ability to break free of the obvious and commonplace, and generate novel ideas and responses [47]. To measure each participant's originality, we first used Agogué and colleagues [41] coding matrix. This matrix, which associates each possible answer to the egg task to a specific score on a 5-point scale ranging from 1 ("not at all original") to 5 ("highly original"), was developed using Amabile's [48] consensual assessment technique. Each participant was then assigned the Top 2 originality index proposed by Silvia and colleagues [49], which consists of the sum of the scores for his/her two most original responses. Following this approach, people are evaluated by the best level of performance they are able to achieve [2] on a constant number of responses. As such, the Top 2 originality index is a better measure of originality than an average score of originality that would penalize respondents who give a large number of uncreative responses. In the end, the participants' individual creativity performance scores were between 1 and 8.8 (mean = 5.48, s.d. = 1.39).

**Participants.** 200 subjects recruited from the online panel of a European professional market research institute provided an informed written consent for experimentation before participating in the study (mean age 41, 51% women). The participants were randomly assigned to one of the four conditions (50 participants per condition). Participants did not differ in terms of age ($F_{(3,196)}$ = .190, *ns*), gender ($\chi^2$ = 1.361, *ns*), education level ($\chi^2$ = 6.085, *ns*), or self-perceived creativity ($F_{(3,196)}$ = 1.058, *ns*) depending on the group they were assigned to.

**Table 1. Pre-label self-perceived creativity items.**

| |
|---|
| I like to discover new ways to consider a problem |
| I like to spend time going beyond the initial perception of a problem |
| I like to take the measure of a situation by considering it as a whole |
| I like working on vaguely-defined or emerging problems |
| I like to use my imagination to generate several ideas |
| I like to work on exceptional ideas |

**Results.** To examine whether the application of a social label as "creative" (or "not creative") influences individual creativity performance, we first conducted an ANOVA considering the manipulation as a between-subjects factor. This ANOVA revealed a significant influence of the manipulation on individual creativity performance ($F_{(3,196)} = 2.927$, $p < .05$, $\eta_p^2 = .043$, $\omega = 0.79$). Corroborating H1, the participants labeled as "creative" performed better in the creative task than those in the control condition (5.81 vs. 5.05, $F_{(1,98)} = 6.884$, $p < .01$, $\eta_p^2 = .066$, $\omega = 0.83$). Also corroborating H2, the participants labeled as "not creative" performed better in the creative task than those in the control condition (5.63 vs. 5.05, $F_{(1,98)} = 4.138$, $p < .05$, $\eta_p^2 = .041$, $\omega = 0.65$). Interestingly, labeled participants' performance in the creative task did not differ according to the positive or negative nature of the label ($F_{(1,98)} = .398$, ns). Also, as expected, the participants labeled as "moderately creative" did not perform differently in the creative task than unlabeled participants in the control condition (5.44 vs. 5.05, $F_{(1,98)} = 2.206$, *ns*). Of note, we observe the same results when controlling for the participants' age, gender, education level, and self-perceived creativity. As a relevant ancillary finding, the analysis also showed that self-perceived creativity does not influence individual creativity performance ($F_{(1,195)} = .026$, *ns*). These results are plotted in Fig 3.

Going further, to test whether self-perceived creativity moderates the influence of a social label marking someone out as creative (or not creative) on individual creativity performance, we conducted a floodlight analysis using the procedure recommended by Cadario and Parguel [50]. Half of the participants were considered in this analysis (50 participants labeled as "creative" and 50 participants labeled as "not creative"). Hayes' [51] PROCESS macro (model 1) and 5000 bootstrapped samples were used to determine whether this moderating effect was significant. The manipulation (i.e. a "not creative" label vs. a "creative" label) was included as the independent variable, and individual creativity performance was included as the dependent variable. Though experimental research scholars often discretize quantitative variables when testing moderation, this practice has been challenged over the last 15 years [52], leading some top-tier journal editors to call for its "death" (e.g. [53]). Participants' self-perceived

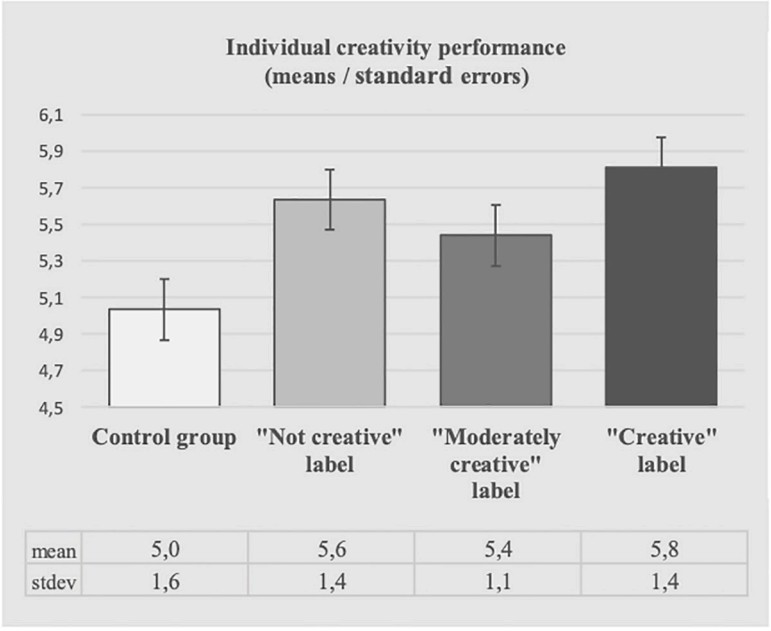

**Fig 3. Results of Study 1.** Direct effects of the labels on individual creativity performance.

creativity was thus included in the floodlight analysis as the moderating variable in its continuous form. This analysis revealed a significant interaction effect between the positive or negative nature of the label and participants' self-perceived creativity ($t_{93}$ = 2.23, $p$ < .05) controlling for participants' age, gender, and education level.

Further conditional analyses show that the "creative" (vs. "not creative") label enhanced individual creativity performance (6.3 vs. 5.2, β = .46, $t_{93}$ = 2.13, $p$ < .05) among the 18 participants scoring relatively low on self-perceived creativity (score under 4 out of 7). It marginally reduced it (5.8 vs. 6.0, β = -.31, $t_{93}$ = 1.38, $p$ < .10) among the 18 participants scoring relatively high on self-perceived creativity (score over 6.33 out of 7). Corroborating H3a but only marginally H3b, these results are depicted in Fig 4.

Study 2 and Study 3 extend these first findings on the impact of social labeling in creativity management by investigating the psychological mediating mechanisms at play when applicating a "creative" label (Study 2) or a "not creative" label (Study 3), versus no label.

## Study 2

Our hypothesis about the effect of labels is based on the idea that labeling an individual alters his or her self-perception. Accordingly, labeling an individual as "creative" should lead to a view of his or her self as creative and result in boosting his or her creative self-efficacy. Fig 5 synthetizes this mediating mechanism.

We now explore the impact of labeling an individual as "creative" (vs. no label) on both his or her self-perceived creativity and creative self-efficacy.

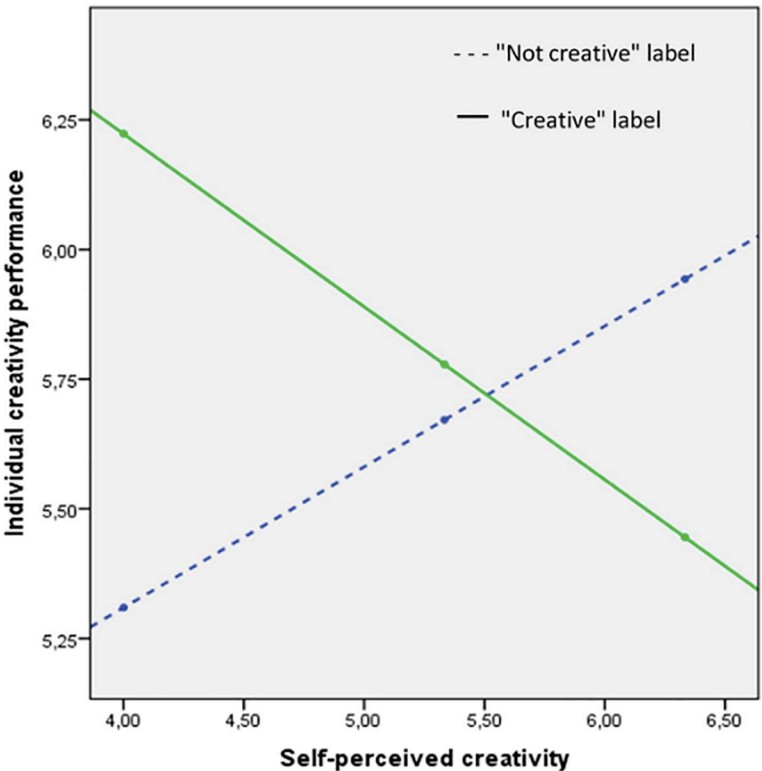

**Fig 4. Moderating effect of self-perceived creativity .**

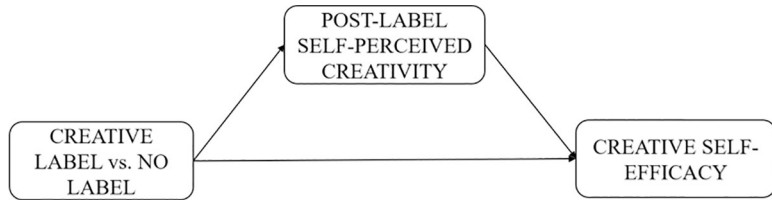

**Fig 5. Conceptual model for Study 2.** Mediating mechanism at play to explain the "creative" label effect.

**Procedure.** Study 2 builds on Study 1 procedure. We first collected self-perceived creativity as a fake basis to ensure effective labeling of participants as "creative", and participants had to register their age, gender and education level. Participants were then randomly assigned to one of the two conditions: a control condition vs. a "creative" label (as manipulated in Study 1). After being randomly exposed to one of the two conditions, they were interrogated about their current self-perceived creativity and about the creative self-efficacy they would feel if they were now confronted with a creative task. To conclude, we asked the participants who were labeled to recall the nature of the label they were provided with to check the validity of our manipulation, verified that no participant correctly guessed the purpose of the study and informed them that the label was purely randomly applied.

**Measures.** *Pre-label self-perceived creativity* was measured using the same 6 items-scale that was initially used in Study 1. To measure *creative self-efficacy*, we expanded Tierney and Farmer's [6] (2002) scale asking respondents how much confident they would feel in their ability if they were "*to participate in a creative exercise now*" using 5 items ranging from 1 "not confident" to 7 "strongly confident": "*ability to address successfully a creative task / Answer creatively to a creative task / Propose many solutions to a creative task / Find out-of-the-ordinary solutions to a creative task / Be original.*" Finally, to measure *post-label self-perceived creativity*, we used a two-item general scale ranging from 1 "not at all " to 7 "very much": "*Right now, you perceive yourself as a creative person / A person who addresses problems in an original way.*"

Our measures proved to be reliable: Cronbach's α of 894 for the items measuring pre-label self-perceived creativity, Cronbach's α of 945 for the items measuring creative self-efficacy and correlation of .794 for the two items measuring post-label self-perceived creativity. Table 2 displays the correlation between all the constructs measuring creative self-perceptions for the two conditions.

**Participants.** 102 subjects recruited from the online panel of a European professional market research institute provided an informed written consent for experimentation before participating in the study (mean age 37, 46% women). The participants were randomly assigned to one of the two conditions (48 participants in the control condition, 54 participants in the "creative" label condition after discarding participants who either didn't fill the questionnaire entirely or couldn't recall correctly how they were labeled). Participants did not differ in terms of age ($F_{(1,101)} = .153$, *ns*), gender ($\chi^2 = 2.387$, *ns*), education level ($\chi^2 = 1.023$, *ns*), or self-perceived creativity pre-label ($F_{(1,74)} = .903$, *ns*) depending on the group they were assigned to.

**Table 2. Correlations.**

|  |  | Pre-label self-perceived creativity | Post-label self-perceived creativity | Post-label creative self-efficacy |
|---|---|---|---|---|
| No label | Pre-label self-perceived creativity | 1 | .514 | .530 |
|  | Post-label self-perceived creativity |  | 1 | .733 |
| "Creative" label | Pre-label self-perceived creativity | 1 | .624 | .713 |
|  | Post-label self-perceived creativity |  | 1 | .822 |

**Table 3. Mediation analysis.**

| | Post label self-perceived creativity | Creative self-efficacy |
|---|---|---|
| No label vs. "Creative" label | **.65**$^*$ | .06 |
| Post-label Self-perceived creativity | | **.78**$^*$ |
| Direct effect of the label on creative self-efficacy IC (95%) | | [-.27;.40] |
| Indirect effect of the label on creative self-efficacy IC (95%) | | **[.13;.91]** |

$^*$ $p < .01$. Results do not change when controlling for participants' pre-label self-perceived creativity.

**Results.**   The aim of this second study is to examine whether the application of a social label as "creative" influences self-perceived creativity and creative self-efficacy. More precisely, Study 2 aims at testing whether self-perceived creativity mediates the influence of a "creative" label (vs. no label) on creative self-efficacy.

To do so, we used Hayes' [51] PROCESS macro (model 4) and 1000 bootstrapped samples. The analysis first shows that the application of the label significantly increases post-label self-perceived creativity (5.2 vs. 4.5 in the "no label" condition, $\beta = .65$, $p < .01$, $\eta_p^2 = .067$, $\omega = 0.85$). Following, post-label self-perceived creativity transfers to creative self-efficacy (5.5 vs. 4.9 in the "no label" condition, $\beta = .78$, $p < .01$). A bias-corrected 95% confidence interval for the indirect effect ranges from 0.13 to 0.91, with a point estimate of 0.51. Since the confidence interval does not include 0, this pattern provides evidence of an indirect effect on post-label self-perceived creativity: the "creative" label enhances self-perceived creativity, which–in turn–enhances creative self-efficacy. This mediation is indirect-only as the direct effect of the "creative" label on creative self-efficacy is not significant ($\beta = .06$, *ns*). Table 3 displays the results of this analysis.

To conclude, Study 2 shows that the application of a "creative" label has the ability to modify creative self-efficacy through self-perceived creativity, which is in line with social labeling theory. In other words, an individual who is labeled as being creative consequently has a higher perception of his or her creativity and therefore has a higher confidence in his or her capacities to perform creatively, explaining why subjects labeled as "creative" were found to display higher creativity performance than unlabeled subjects in Study 1.

## Study 3

Our hypothesis about the effect of the "not creative" label is based on the idea that labeling an individual alters his or her self-perception. Accordingly, labeling an individual as "not creative" should both lead to a view of his or her self as not creative and encourage him or her to invest efforts to display high creativity performance in a subsequent creative task in order to restore it. Fig 6 synthetizes this mediating mechanism.

To test these mediating mechanisms, we explore the impact of labeling an individual as "not creative" (vs. no label) on his or her self-perceived creativity, involvement in the creative task and individual creativity performance.

**Procedure.**   Study 3 builds on Study 1 and Study 2 procedure. We first collected self-perceived creativity as a fake basis to ensure effective labeling of participants as "not creative", and participants had to register their age, gender and education level. Participants were then randomly assigned to one of the two conditions: a control condition vs. a "not creative" label. We then interrogated the participants about their current self-perceived creativity and invited them to participate in a creative task–to come-up with creative ideas for a brick (no minimal time). To conclude, we asked the participants who were labeled to recall the nature of the label they were provided with to check the validity of our manipulation, verified that no participant

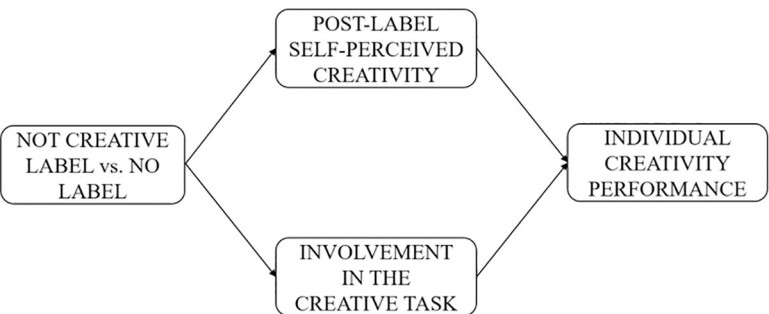

**Fig 6. Conceptual model for Study 3.** Mediating mechanism at play to explain the "not creative" label effect.

correctly guessed the purpose of the study and informed them that the label was purely randomly applied.

**Measures.** *Pre-label self-perceived creativity* was measured using the same 6 items-scale that was used in Study 1 and Study 2 (Cronbach's α of 899). To measure *post-label self-perceived creativity*, we also used the same 2 items-scale that was used in Study 2 (correlation of .731). To measure *involvement in the creative task*, we monitored the time spent on the task. Finally, individual creativity performance was measured, as in Study 1, using Silvia and colleague's Top 2 originality index [50]. Its scores were between 0 and 14 (mean = 5.34, s.d. = 3.46).

**Participants.** Subjects recruited from the online panel of a European professional market research institute provided an informed written consent for experimentation before participating in the study (mean age 42, 58% women). They were randomly assigned to one of the two conditions. 180 participants were considered in the analyses (108 participants in the control condition, 72 participants in the "not creative" label condition after discarding participants who either didn't fill the questionnaire entirely, didn't produce any idea or couldn't recall correctly how they were labeled). Participants did not differ in terms of age ($F_{(1,178)}$ = .420, *ns*), gender ($\chi^2$ = .034, *ns*), education level ($\chi^2$ = 6.323, *ns*), or self-perceived creativity pre-label ($F_{(1,178)}$ = .036, *ns*) depending on the group they were assigned to.

**Results.** The aim of Study 3 is to examine whether self-perceived creativity and the involvement in the creative task mediate the influence of a "not creative" label on individual performance creativity. To do so, we used Hayes' [51] PROCESS macro (model 4) and 1000 bootstrapped samples. The analysis first shows that the application of the "not creative" label significantly reduces post-label self-perceived creativity (4.4 vs. 4.9 in the "no label" condition, β = -.59, p < .01, $\eta_p^2$ = .059, ω = 0.92) and increases time spent on the task (138 vs. 93 seconds in the "no label" condition, β = 51.77, p < .05, $\eta_p^2$ = .018, ω = 0.43). Following, both post-label self-perceived creativity (β = .34, *p* < .05) and time spent on the task (β = .0058, *p* < .01) transfer to individual performance creativity, which scores at 5.7 for the participants labeled as "not creative" vs. 5.1 for the unlabeled participants. A bias-corrected 90% confidence interval for the indirect effect ranges from -0.42 to -0.02, with a point estimate of -0.20 for post-label self-perceived creativity and from 0.04 to 0.56, with a point estimate of 0.30 for the time spent on the task. Since the confidence interval does not include 0, this pattern provides evidence of a marginally significant indirect effect on individual performance creativity: the "not creative" label reduces self-perceived creativity but enhances the involvement in the creative task, which–in turn–modifies individual performance creativity. This mediation is partial as the direct effect of the "not creative" label on individual performance creativity is also marginally significant (β = .76, *p* < .10). Table 4 displays the results of this analysis.

To conclude, Study 3 shows that the application of a "not creative" label damages self-perceived creativity and enhances involvement in the creative task. As the second mechanism

**Table 4. Mediation analysis.**

| | Post label self-perceived creativity | Involvement in the creative task | Individual performance creativity |
|---|---|---|---|
| No label vs. "Not creative" label | -.59** | 51,77* | |
| Post-label Self-perceived creativity | | | .34* |
| Involvement in the creative task | | | .0058** |
| Direct effect of the label on individual performance creativity IC (90%) | | | [.02;1.51] |
| Indirect effect via post-label self-perceived creativity IC (90%) | | | [-.42;-.02] |
| Indirect effect via the involvement in the creative task IC (90%) | | | [.04;.56] |

** *p* < .01

* *p* < .05. Of note, when controlling for participants' pre-label self-perceived creativity, the indirect effect via post-label self-perceived creativity is no more significant.

appears stronger than the first one, which disappears when controlling for pre-label self-perceived creativity, this could explain why, on the whole, subjects labeled as "not creative" displayed higher creativity performance than unlabeled subjects ($\beta$ = .87, p < .05, $\eta_p^2$ = .019, $\omega$ = 0.46), corroborating Study 1 findings.

## Discussion

In this paper, we explore the relevance of social labeling techniques to foster individual creative behaviour. We show that labeling collaborators as "creative" or "not creative" has the potential to enhance their individual creativity performance, and that the effectiveness of labeling depends on the way individuals initially perceive themselves in terms of creativity. These findings make a number of relevant contributions for both academics and practiners involved in innovation management.

### Theoretical contributions

This research contributes to the literature on social labeling. In showing the positive effect of a positive "creative" label on creative behaviors, it replicates previous findings observed in psychology and social marketing in an innovation management setting, thus extending the boundaries of social labeling's relevance to the organizational field. In demonstrating the moderating effect of self-perceived creativity, it also offers a theoretical validation of social labeling theory itself, whose basic mechanism involves an alteration of individuals' self-perception. Furthermore, in showing the positive effect of a negative "not creative" label, it corroborates the congruence hypothesis formulated by DeJong [35] and Guéguen [25], according to which a negative label only generates reactance when the label and the requested behavior are unconnected, such that the requested behavior offers no relevant opportunity to contradict the negative label. Additionnaly, Study 2 and Study 3 provide convincing empirical evidence about the mechanism at play behind the effect of each type of label.

Though demonstrating the significant positive effect of the "not creative" label on individual creativity performance, Study 1 and Study 3 also display specific marginal effects to be discussed. In Study 1, an insufficient sample size when testing the moderating influence of self-perceived creativity may be responsible for this result. Still, Study 3 has provided additional data showing that the application of a "not creative" label enhances subjects involvement in the creative task, but only among those scoring high on self-perceived creativity. This external evidence (not presented here to keep the paper consistent and reasonable in size) adds confidence in the moderating influence of self-perceived creativity in the case of the "not creative" label. In Study 3, the marginal effect we found applies to the mediation by time spent on the

creative task. As behavioral effects are often more challenging to study, demonstrating the mediating role of subjects' involvement in the influence of the "not creative" label on creative behavior already provides a relevant finding. In the end, the marginal effects we found in Study 1 and Study 3 do not undermine our general and pioneering contribution while still calling for further research to more finely understand social labeling techniques.

Beyond, this research also contributes to the literature on creativity. First, it shows the efficiency of operational devices using self-perception mechanisms to avoid forms of psychological reactance that could negatively affect individual creativity performance [22]. Positive labels, far from being perceived as an external influence, are received as the external recognition of the individual's internal dispositions, and as such do not generate reactance. Second, this research sheds light on the psychological foundations of the Pygmalion process described by Tierney and Farmer [7], where creative self-efficacy mediates the effects of supervisor expectations on individual creativity performance. Third, this research also offers insights on using social labeling as a more frugal and less invasive way for managers to trigger creative self-efficacy than existing procedures (e.g. [6]). Social labeling techniques do not require managers to engage in any concrete creative process, nor to be creative themselves. This research thus contributes to the existing literature on the role of leadership for creativity, investigating minimal-resource strategies to increase individual creativity performance [54] [55]. It also calls for further research on the potential of nudges [56] in creativity management, since direct stimulation of creativity such as feedbacks from managers has been proven to have mixed results [45] [57].

Last, from the perspective of the innovation management literature, this research helps understand the "champion" phenomenon, in which individual intrapreneurs are recognized as developing unusual skills of innovation in their organizations [58] [59]. Such people may perceive themselves as highly creative, but are labeled as "not creative" by being assigned to operational teams: rather than losing self-esteem and work motivation and underperforming, they "overperform" to contradict the label [4] [60]. Labels already exist in most organizations. Large organizations, even industrial entities and institutions often implicitly convey positive stereotypes about specific members such as industrial designers [61], bioinformaticians [62] or web-designers and visual artists [63], and negative stereotypes about other members (e.g. buyers, lawyers or accountants). Establishing routines and processes highlights this separation between creativity and operational decision-making [64]. Further research should explore whether implicit "creative" and "not creative" labels have the same effects as explicit ones.

## Managerial implications

First of all, our findings have implications for directly managing individual creativity in one-to-one settings between a manager and his or her employees. Creativity is a key factor in exploratory development of disruptive ideas [65] and innovation in highly competitive markets [66]. As such, the ability to boost employees' creativity is more and more expected from managers. As social labeling appears as a frugal tool to influence behaviours, our findings suggest that such techniques can be used to foster creativity in situations where individual creativity is expected but does not benefit from much structural support (e.g. in small organizations such as start-ups or SMEs, or whenever an urgent creative answer is required, even in large companies).

Yet, social labels need to be credible to mask any persuasive intent and be effective [9]. This raises the question of how a "creative" or "not creative" label could be credible in an organization. Praising creative efforts, even if they are unsuccessful, has been proposed by Tierney and Farmer [7], but may lack credibility unless it is backed up by a record of previous

achievements. Other sources of plausibility may lie in the reference to past achievements or in the identity of the label giver. Ideally, the label giver should be legitimate: this does not mean that he or she needs to display tremendous creativity, but that he or she has recognized skills in terms of creativity evaluation and selection. Organizational structure, and its implicit labeling system, could be perceived as more credible than any HR or communication department, which would not be recognized as a legitimate label giver. Being impersonal in nature, processes should be under less suspicion of persuasive intent than a manager, whose personal approaches can always be questioned.

Interpreting the paradoxical effects of positive and negative labels on creativity, several managerial recommendations finally result from our research. The first one is to remind operational employees–who may not perceive themselves as being creative in their occupation–that they too are creative, if innovative behaviors are to be expected subsequently. Low scorers on self-perceived creativity will then perform better in terms of creativity. On the other hand, for individuals having a high belief in their own creativity, psychological reactance is bounded when a negative label is applied, and our research suggests that the "not creative" label should be applied to them to elicit the desired creative behavior [25] [35]. One may object that such a managerial act may cause relation conflicts within a team. However, the study by Jung and Lee [67] showed that for invidiuals who value harmonious relationships in the workplace, conflictual relationalship may contrigger the perception of the problematic situation from a new standpoint and harness individual creative behavior to overcome the situation. This clearly calls to further explore the role of negative labels on creativity, integrating moderating factors such as placing great value on creating hamonious relationships.

## Limitations and further research

The present research is not without limitations. First, the nature of the label we used was quite simple: we used labels that relate to the level of creativity of individuals in a broad sense, by labeling individuals as "creative", "moderately creative" or "not creative". Building on the specific cognitive mechanisms inherent to creativity, such as associative thinking [68], further research could thus investigate the influence of more elaborate social labeling, specifically focusing on the individual ability to perform one particular cognitive mechanism. For instance, building on the very recent concept of idea linking that has been shown to stimulate creativity, labels such as "*you are a person who is very good at using aspects of early ideas as input for subsequent ideas in a sequential manner*" could provide fruitful insights on the managerial levers to boost individual creativity through social labeling.

Second, our experimental protocol was built on a label automatically provided by the computer to our participants, leaving a blind spot regarding the nature of the existing relationship between managers and employees in real-life settings. Indeed, leader-member exchange has been consistently shown to positively influence creativity [69], specifically when creative self-efficacy mechanisms are at stake [70], calling for further investigation regarding the moderating influence of leader-member relationships on the effectiveness of social labeling for creativity.

Third, we didn't test the persistence of the different labels' effects: we measured the influence of labels on self-perceived creativity, creative-self efficacy, and creativity performance just after applying a label but did not examine its lasting influence. As such, further research could investigate the long-lasting influence of social labeling on creativity, specifically in the case of negative labels that may, in the long-term term, influence work identity and task motivation.

Last, we should stress that we investigated only one part of the creative process, namely individual idea generation, calling to extend our findings both on the consequent phases of the creative process, namely idea evaluation and selection, and on more collective settings.

## Supporting information

**S1 File. Data for Study 1.**
(SAV)

**S2 File. Data for Study 2.**
(SAV)

**S3 File. Data for Study 3.**
(SAV)

## Acknowledgments

The authors would like to thank Sophie Hooge, Mathieu Cassotti and Denis Grégoire for their insights on previous versions of this paper, as well as the editor Valerio Capraro and the reviewers for their insightful comments on previous versions of the paper.

## Author Contributions

**Conceptualization:** Marine Agogué, Béatrice Parguel.

**Methodology:** Marine Agogué, Béatrice Parguel.

**Validation:** Marine Agogué, Béatrice Parguel.

**Writing – original draft:** Marine Agogué, Béatrice Parguel.

**Writing – review & editing:** Marine Agogué, Béatrice Parguel.

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
