## [Decision Letter · Decision Letter 0]

24 Jul 2019

PONE-D-19-16350

Nudging individuals’ creativity using social labeling : an experimental study

PLOS ONE

Dear Dr Agogué,

Thank you for submitting your manuscript to PLOS ONE. After careful consideration, we feel that it has merit but does not fully meet PLOS ONE’s publication criteria as it currently stands. Therefore, we invite you to submit a revised version of the manuscript that addresses the points raised during the review process.

Please find the reviewers' comments below.

We would appreciate receiving your revised manuscript by Sep 07 2019 11:59PM. To enhance the reproducibility of your results, we recommend that if applicable you deposit your laboratory protocols in protocols.io, where a protocol can be assigned its own identifier (DOI) such that it can be cited independently in the future. For instructions see: http://journals.plos.org/plosone/s/submission-guidelines#loc-laboratory-protocols

We look forward to receiving your revised manuscript.

Kind regards,

Valerio Capraro

Academic Editor

PLOS ONE

Journal Requirements:

Additional Editor Comments (if provided):

I have now collected two reviews from two experts in the field. Both reviewers recommend major revisions. Therefore, I would like to invite you to revise the paper according to the reviewers' suggestions. Needless to say that all comments must be addressed. Particular attention, however, should be given to the issue raised by both reviewers regarding the smallness of the sample (especially in light of the current replicability crisis) and to the issue raised by Reviewer 1 regarding deception (please explain exactly which parts of the experiments involved deception and why you think this is not a problem for your experiments). Moreover, I would like to mention that social labels have been recently used also to impact people's decisions in economic games (https://www.sciencedirect.com/science/article/pii/S0022103116302098;
https://www.sciencedirect.com/science/article/pii/S0022103118302841;
http://journal.sjdm.org/17/171107/jdm171107.pdf;
http://journal.sjdm.org/19/190107/jdm190107.pdf). Note that I am the author of some of these papers. My decision on this manuscript will obviously be independent of whether you will decide to include or not these papers in your reference list. I have just thought that you might find them relevant.

Looking forward for the revision.

Reviewers' comments:

Reviewer's Responses to Questions

**Comments to the Author**

1. Is the manuscript technically sound, and do the data support the conclusions?

Reviewer #1: Partly

Reviewer #2: Partly

2. Has the statistical analysis been performed appropriately and rigorously? 

Reviewer #1: Yes

Reviewer #2: No

3. Have the authors made all data underlying the findings in their manuscript fully available?

Reviewer #1: Yes

Reviewer #2: Yes

4. Is the manuscript presented in an intelligible fashion and written in standard English?

Reviewer #1: Yes

Reviewer #2: Yes

5. Review Comments to the Author

Reviewer #1: Did you do a power test to determine whether your 76-subject study n was large enough to be statistically representative of the population? 76 people does not sound like a big sample.

Can you explain what alpha means in the following sentences?

Our measures display good reliability indicators: Cronbach’s α of 874 for the items measuring pre-label self-perceived creativity, Cronbach’s α of 941 for the items measuring creative self-efficacy and correlation of .803 for the two items measuring post-label self- perceived creativity.

I am not sure about PLOS-One, but I know that most experimental economics journals does not allow for deception. You clearly used deception in this experiment by randomly assigning people to the “creative” and “non-creative” groups. If PLOS-One is fine with deception, then this is not an issue. However, it is a big issue with experimental economists.

I would like to see demographic and other sample characteristics for the two samples of the control vs. treatment for each study. Are there any demographic differences? With a total n of 76, I would like to know if there are differences other than just the creativity score.

Again, with study 2, did you do a power test on your n=200 participants to see if this was a sufficient number of subjects?

Again, can you show the demographic characteristics of the control vs treatments for study 2?

Reviewer #2: This study explores whether social labels could influence creative behavior.

Major issues:

In the introduction, researchers suggested that “a labeled individual appears to internalize the values or personality traits associated with the label.” Later, authors argue that for negative labels there is a different mechanism “participants could re-attribute a negative label to their negative qualities, and make efforts to disprove it” As it stands now, it seems that labels influence behavior via different mechanisms, internalization, and resistance to the negative label. While it could be the case, neither study 1 nor study 2 directly measure internalization or resistance to internalize the label. Thus, overall, there is no compelling case that empirically demonstrates either of mechanisms.

I could argue that the internalization happens in both cases: positive label increases self-perception and individuals perform better; negative label decreases self-perception of creativity that makes individuals work harder. (you suggested similar mechanism explaining the moderation later in the paper) If this explanation is the case, there need to be a study that demonstrates this effect. Namely, negative labeling decreases self-perception and that leads to better performance, positive labeling increases self-perception and that leads to better performance than in the control condition.

- The section “Boosting individual creativity within organizations” does not seem to contribute to developing the story. Neither organizational structures nor supervisor advice is studied in this research. I suggest removing or reorganizing these sections in a way that includes explicit connections to the goals and measures of the study.

- The authors do not provide enough evidence that supports the following statement “the application of a “creative” label will be effective if the subjects do not perceive the persuasion attempt behind the label.” Why individuals who perceive the persuasion attempt behind the label will not be empowered by it? There is literature suggesting that if people know about persuasive effects of nudges they are still affected by them (e.g. Loewenstein G, Bryce C, Hagmann D, Rajpal S. Warning: You are about to be nudged.)

- It is possible that “a “creative” label enhances self-perception of creativity while a “not creative” label erodes self-perception. The moderation analysis highlights this possibility. However, to have evidence for this statement, it is important to incorporate the measures of self-perception before and after the manipulation to demonstrate that with a positive label the self-perception goes up, while with “negative label” self-perception goes down.

- Study 1

- Samples size in study one is 76 people which suggests that the study is probably underpowered. Did you do power calculation and a sample size calculation?

- There was randomization in this study, why age, gender, education level, and self-perceived creativity are added as controls? There was no difference in these measures between groups, therefore these controls should be removed from the analysis

- The goal of the study was to show that the label influences participants’ self-perception of creativity, which was achieved. Why there was an analysis of self-efficacy? If it highlights the mechanism of how labels influence performance, why there is no measure of self-efficacy in Study 2? I would not state in the discussion that this research contributes to self-efficacy literature as there is no association between self-efficacy and the study conditions or performance.

- Neither design of this study nor analysis speaks to this hypothesis. While this study could be a part of the research package, it does not directly speak to any of the stated hypothesizes. The measure of self-perception demonstrates rather that the manipulation worked as researchers planned it.

Study 2

- Where the participants' removal in this study? Did you have the same manipulation checks as in the previous one? Did participants remember the labels correctly?

- Covariates should not be included in the experimental study, especially that there were no differences in these variables between conditions.

- Please report degrees of freedom in moderation study, it seems underpowered, especially for the continuous measure.

- While moderation analysis demonstrates the possibility of the highlighted in the introductions relationships, the further mediation analysis is needed to provide further evidence (see my comments above)

- I don’t see the value in following Hayes process with ANCOVA, you could remove it from the study and visualize your observations directly from the Process Model 1 procedure.

OVERALL

- I recommend re-organizing the description of the mechanism/s via which positive and negative labels influence behavior.

- Recommend additional study that has exact measures to test the proposed mechanisms (see my comments above)

Minor issues

- While I can see that the work of Bem, 1972 might be related to social labeling, It would be great if authors could elaborate on the connection between self-perception theory and social labeling.

- I am not sure what you mean here “In facts, for the self-perception to be altered, the persuasive intent behind the label must be masked, explaining why experiments often rely on people’s previous statements…”

6. PLOS authors have the option to publish the peer review history of their article (what does this mean?). If published, this will include your full peer review and any attached files.

Reviewer #1: No

Reviewer #2: No

---

## [Author Response · Author response to Decision Letter 0]

1 Oct 2019

Nudging individuals’ creativity using social labeling: an experimental study

Manuscript: PONE-D-19-16350

Reviewer #1

Dear Reviewer #1,

We sincerely thank you for your very detailed and relevant comments on our manuscript. 

To facilitate your review of our answers and changes, we discuss your comments and explain how we have integrated each of them into this new version of our manuscript directly below each comment. Your original comments are reproduced in bold; our responses follow each comment; the modified text appearing in the new version of our manuscript is copied in blue.

Did you do a power test to determine whether your 76-subject study sample was large enough to be statistically representative of the population? 76 people does not sound like a big sample.

We thank you for this comment. Originally educated in psychology, we more often use the criterium of the effect size to evaluate whether results are important enough. However, we do acknowledge that the power of the test appears today as a more robust criterium. 

To better satisfy the power criterium, we collected more data to complete Study 1 sample (please note that we now present Study 1 as Study 2). Precisely, we asked our market research institute to propose our study to 30 more panelists. These panelists were exposed to the “creative” label condition, which was previously underrepresented (n=28 versus n=48 for the control condition). Out of these 30 more respondents, 26 checked the manipulation regarding the label and were included in the database we provide along with our new submission (these new respondents are identified as 101, 102… 126 in the database). Following, the section dedicated to the presentation of the results of the study has been fully rewritten in our new submission to consider our data collection extension. Precisely, it shows that our results stay the same but that the addition of respondents allowed an increase in the power of our test, which reaches now 84,5% as now explicitly appears in the manuscript:

To do so, we used Hayes' (2012) PROCESS macro (model 4) and 1000 bootstrapped samples. The analysis first shows that the application of the label significantly influences post-label self-perceived creativity (�=.65, p<.01, ηp2=.067, ω=0.85).

Can you explain what alpha means in the following sentences? “Our measures display good reliability indicators: Cronbach’s α of 874 for the items measuring pre-label self-perceived creativity, Cronbach’s α of 941 for the items measuring creative self-efficacy and correlation of .803 for the two items measuring post-label self-perceived creativity.”

The Cronbach’s alpha measures internal consistency, i.e. how closely related a set of items are as a group. It is a measure of scale reliability. As the Cronbach’s alpha is quite a classical indicator in statistical test theory, we do not propose to add more explanation about what it means in the manuscript. If we did not understand your comment properly, please be more specific, we will be more than happy to answer in a more appropriate way. 

I am not sure about PLOS-One, but I know that most experimental economics journals does not allow for deception. You clearly used deception in this experiment by randomly assigning people to the “creative” and “non-creative” groups. If PLOS-One is fine with deception, then this is not an issue. However, it is a big issue with experimental economists.

You are right mentioning that we designed our experiment as a deception experiment and also perfectly right mentioning that deception is a big issue in experimental economics. As mentioned by Tyler and Amodio in 2015 the Oxford Handbook of Experimental Economic Methodology, “there is no subject that provokes so much emotion among economists as the use of deception in an experiment”.

Tyler, T. R., & Amodio, D. M. (2015). Psychology and economics: areas of convergence and difference. Handbook of experimental economic methodology. Oxford University Press, Oxford, 181-196.

Still, deception experiments are more than frequent in psychology experiments and have been regarded as legitimate by many social sciences that have been using experiments for decades.

Deception experiments are not used by psychologist for amoral reasons, but to ensure internal validity. Precisely, our manipulation allows to draw conclusion that could not be drawn otherwise. We can not imagine that it would have been possible to tell our subjects, as an introduction of the experiment, that we were about to put them in different fake conditions regarding their personal creativity. Such a disclosure would clearly have modified the way our subjects would have presented themselves and the way they would have responded to our label.

Tyler and Amodio (2015) identify a specific problem with deception experiments as they can “spoil the subject pool” for subsequent experiments. Though this argument could be relevant when carrying experiments using internal student samples, it is less relevant when considering the external sample of a market research institute. As a matter of fact, we have been using the same market research institute for years and always ask our subjects what they thought was the actual objective of the study they had just participated in and none of them (in all our data collections) correctly guessed it. Following, we are pretty confident on the fact that deception experiments conducted on very large pool of respondents and not on very restricted local pool do not “spoil the sample”.

Another problem is more moral and regards the fact to put subjects in scenario that make them believe for few minutes that they are not creative. To circumvent this problem, we used a clear debriefing statement at the end of the experiment to make clear that all subjects are. As an addition, we would also like to mention that all subjects participating in our studies gave an active consent before participating and that our protocols were all validated by one of the researchers’ IRB.

A written in the first version of our manuscript:

For ethical purposes, after completing the whole procedure, participants were informed in the following words that the label was purely randomly applied: “Your creativity score stated at the start of the survey was randomly assigned for the purpose of our research. It does not reflect your actual level of creativity at all.” As for Study 2 and Study 3, Study 1 experimental procedure was approved by the first author’s Institutional Review Board.

In the new version, we add a specific comment on deception (see the sentence in bold characters in the previous paragraph) to fully address your comment.

Of note, we acknowledge that disguising what we were studying deceives subjects but appears necessary to preserve the internal validity of our results. Though it can be considered as a big issue by economists, most social sciences that have been carrying experiments for decades tend to consider it as a legitimate procedure (Tyler & Amodio, 2015).

I would like to see demographic and other sample characteristics for the two samples of the control vs. treatment for each study. Are there any demographic differences? With a total n of 76, I would like to know if there are differences other than just the creativity score.

We provide statistics about demographic differences as follows in Study 1:

102 subjects recruited from the online panel of a European professional market research institute provided an informed consent for experimentation before participating in the study (mean age 37, 46% women). The participants were randomly assigned to one of the two conditions (48 participants in the control condition, 54 participants in the “creative” label condition after discarding participants who either didn’t fill the questionnaire entirely or couldn’t recall correctly how they were labeled). Participants did not differ in terms of age (F(1,101)=.153, ns), gender (χ²=2.387, ns), education level (χ²=1.023, ns), or self-perceived creativity pre-label (F(1,74)=.903, ns) depending on the group they were assigned to.

Again, with study 2, did you do a power test on your n=200 participants to see if this was a sufficient number of subjects?

For Study 2 (now Study 1), the tests we carried show that the size of our effects are between medium and large. The power of our tests mostly appears over .70 and for H1 over .80, as detailed in the new version of our manuscript:

To examine whether the application of a social label as “creative” (or “not creative”) influences individual creativity performance, we first conducted an ANOVA considering the manipulation as a between-subjects factor. This ANOVA revealed a significant influence of the manipulation on individual creativity performance (F(3,196)=2.927, p<.05, ηp2=.043, ω=0.79). Corroborating H1, the participants labeled as “creative” performed better in the creative task than those in the control condition (5.81 vs. 5.05, F(1,98)=6.884, p<.01, ηp2=.066, ω=0.83). Also corroborating H2, the participants labeled as “not creative” performed better in the creative task than those in the control condition (5.63 vs. 5.05, F(1,98)=4.138, p<.05, ηp2=.041, ω=0.65). 

We also found a power over .70 for our floodlight analysis as shown hereafter:

To make this interaction effect easier to communicate (Fitzsimons, 2008), we dichotomized participants between higher and lower scorers on pre-label self-perceived creativity, using a median-split. We then carried out a classic ANOVA considering the manipulation (i.e. a “not creative” label vs. a “creative” label) and the dichotomized version of pre-label self-perceived creativity as between-subjects factors. This ANOVA replicated the significant interaction effect previously identified by the floodlight analysis between the positive/negative nature of the label and participants’ self-perceived creativity (F(1,96)=4.898, p<.05, ηp2=.049, ω=0.71). Planned contrast tests also revealed that the “creative” label enhanced individual creativity performance among participants scoring low on self-perceived creativity, who outperformed the “not creative” label group (6.18 vs. 5.45, F(1,49)=6.417, p<.01, ηp2=.116, ω=0.80). However, among participants scoring high on self-perceived creativity, the “not creative” group outperformed the “creative” label group, though in a marginally significant way (5.90 vs. 5.41, F(1,41)=2.387, p<.10, ηp2=.055, ω=0.45). Corroborating H3a but only marginally H3b, these results are depicted in Figure 4.

Again, can you show the demographic characteristics of the control vs treatments for study 2?

We provide statistics about demographic differences as follows in Study 2:

200 subjects recruited from the online panel of a European professional market research institute provided an informed consent for experimentation before participating in the study (mean age 41, 51% women). The participants were randomly assigned to one of the four conditions (50 participants per condition). Participants did not differ in terms of age (F(3,196)=.190, ns), gender (χ²=1.361, ns), education level (χ²=6.085, ns), or self-perceived creativity (F(3,196)=1.058, ns) depending on the group they were assigned to.

As we added a study 3, we provide statistics about demographic differences as well for this third story. 

 

Reviewer #2

Dear Reviewer #2,

We sincerely thank you for your very detailed and relevant comments on our manuscript. 

To facilitate your review of our answers and changes, we discuss your comments and explain how we have integrated each of them into this new version of our manuscript directly below each comment. Your original comments are reproduced in bold; our responses follow each comment; the modified text appearing in the new version of our manuscript is copied in blue.

In the introduction, researchers suggested that “a labeled individual appears to internalize the values or personality traits associated with the label.” Later, authors argue that for negative labels there is a different mechanism “participants could re-attribute a negative label to their negative qualities, and make efforts to disprove it” As it stands now, it seems that labels influence behavior via different mechanisms, internalization, and resistance to the negative label. While it could be the case, neither study 1 nor study 2 directly measure internalization or resistance to internalize the label. Thus, overall, there is no compelling case that empirically demonstrates either of mechanisms. I could argue that the internalization happens in both cases: positive label increases self-perception and individuals perform better; negative label decreases self-perception of creativity that makes individuals work harder. (you suggested similar mechanism explaining the moderation later in the paper) If this explanation is the case, there need to be a study that demonstrates this effect. Namely, negative labeling decreases self-perception and that leads to better performance, positive labeling increases self-perception and that leads to better performance than in the control condition.

We sincerely thank you for this comment that appears perfectly legitimate and invited us to enhance our argument and the quality of our manuscript. 

In the new version that we are submitting, you will find a whole new data collection, which focuses on the influence of the “not creative” label and demonstrates that when labeled as “not creative”, subjects scoring high on self-perceived creativity display a lower perception of their creativity and an increase in their involvement in the creative task, as measured by the time spent on the task. Please see below:

Study 3

Our hypothesis about the effect of the “not creative” label is based on the idea that labeling an individual alters his or her self-perception. Accordingly, labeling an individual as “not creative” should lead to a view of his or her self as not creative and encourage him or her to invest efforts to display high creativity performance in a subsequent creative task in order to restore it. To test this mediating mechanism, we explore the impact of labeling an individual as “not creative” (vs. no label) on both his or her self-perceived creativity and involvement in the creative task.

Procedure

Study 3 builds on Study 1 and Study 2 procedure. We first collected self-perceived creativity as a fake basis to ensure effective labeling of participants as “not creative”, and participants had to register their age, gender and education level. Participants were then randomly assigned to one of the two conditions: a control condition vs. a “not creative” label. We then interrogated the participants about their current self-perceived creativity and invited them to participate in a fake creative task. To conclude, we asked the participants who were labeled to recall the nature of the label they were provided with to check the validity of our manipulation, verified that no participant correctly guessed the purpose of the study and informed them that the label was purely randomly applied.

Measures

Pre-label self-perceived creativity was measured using the same 6 items-scale that was used in Study 1 and Study 2 (Cronbach’s α of 899). To measure post-label self-perceived creativity, we also used the same 2 items-scale that was used in Study 2 (correlation of .731). Finally, to measure involvement in the creative task, we monitored the time spent on the task.

Participants

202 subjects recruited from the online panel of a European professional market research institute provided an informed consent for experimentation before participating in the study (mean age 42, 58% women). The participants were randomly assigned to one of the two conditions (120 participants in the control condition, 82 participants in the “not creative” label condition after discarding participants who either didn’t fill the questionnaire entirely or couldn’t recall correctly how they were labeled). Participants did not differ in terms of age (F(1,200)=.620, ns), gender (χ²=.021, ns), education level (χ²=5.315, ns), or self-perceived creativity pre-label (F(1,200)=.007, ns) depending on the group they were assigned to.

Results

The aim of Study 3 is to examine whether self-perceived creativity mediates the influence of a “not creative” label on the involvement in the creative task. To do so, we used Hayes' (2012) PROCESS macro (model 4) and 1000 bootstrapped samples but did not find any significant mediation between the construct. However, and as expected, the analysis shows that the application of the “not creative” label (vs. no label) both significantly damages post-label self-perceived creativity (F=12.19, p<.01, ηp2=.057, ω=0.97) and increases the time spent on the task (F=3.03, p<.05, ηp2=.015, ω=0.54).

Going further, to test whether the influence of the “not creative” label (vs. no label) on participants’ involvement in the creative task could depend on pre-label self-perceived creativity, as suggested by findings from Study 1, we conducted a floodlight analysis. Hayes' (2012) PROCESS macro (model 1) and 5000 bootstrapped samples were used to determine whether this moderating effect was significant. The manipulation (i.e. a “not creative” label vs. no label) was included as the independent variable, the time spent on the task as the dependent variable, and participants’ self-perceived creativity in its continuous form as the moderating variable. This analysis revealed a significant interaction effect between the manipulation and participants’ self-perceived creativity (t=1.98, p<.05).

As in Study 1, to make this interaction effect easier to communicate, we dichotomized participants between higher and lower scorers on pre-label self-perceived creativity, using a split at the Johnson-Neyman point (i.e. over 5.50). We then carried out a classic ANOVA considering the manipulation (i.e. a “not creative” label vs. no label) and the dichotomized version of pre-label self-perceived creativity as between-subjects factors. This ANOVA replicated the significant interaction effect previously identified by the floodlight analysis (F(1,198)=4.845, p<.05, ηp2=.024, ω=0.71). Planned contrast tests then revealed that, among participants scoring high on self-perceived creativity, the “not creative” label (vs. no label) significantly enhanced their involvement in the creative task (208’ vs. 97’, F(1,78)=3.804, p<.05, ηp2=.047, ω=0.61). No such influence appeared among participants scoring low on self-perceived creativity (88’ vs. 91’, F(1,120)=.017, ns).

To conclude, Study 3 shows that the application of a “not creative” label damages self-perceived creativity of subjects scoring high on self-perceived creativity and enhances their involvement in the creative task. Among subjects scoring low on self-perceived creativity, it only damages self-perceived creativity. These results could explain why subjects labeled as “not creative” were found to be likely to display higher creativity performance than unlabeled subjects in Study 1.

The section “Boosting individual creativity within organizations” does not seem to contribute to developing the story. Neither organizational structures nor supervisor advice is studied in this research. I suggest removing or reorganizing these sections in a way that includes explicit connections to the goals and measures of the study.

We agree with this observation. We have reframed and tightened the literature section by focusing specifically and direction and indirect stimuli that managers can use to trigger creative behavior. 

The authors do not provide enough evidence that supports the following statement “the application of a “creative” label will be effective if the subjects do not perceive the persuasion attempt behind the label.” Why individuals who perceive the persuasion attempt behind the label will not be empowered by it? There is literature suggesting that if people know about persuasive effects of nudges they are still affected by them (e.g. Loewenstein G, Bryce C, Hagmann D, Rajpal S. Warning: You are about to be nudged.)

I am not sure what you mean here “In facts, for the self-perception to be altered, the persuasive intent behind the label must be masked, explaining why experiments often rely on people’s previous statements…”

Thank you for this comment. We really enjoyed the reading of the article you mention about the effects of disclosing the presence of default options and discussed a lot about it.

In the end, we retain the following conclusion made by Loewenstein and colleagues (2015): “our findings demonstrate that default options are a category of nudges that can have an effect even when people are aware that they are in play” (p. 40). It is actually interesting to consider that different categories of nudges could have different effects when they are disclosed. As far as we are concerned in the present manuscript, our nudge is based on self-perception theory and implies deception, which is quite common and legitimate in experimental psychology. Following, disclosing our manipulation would have meant to mention right from the beginning to our subjects that we were about to lie to them when labeling them as “x” or “Y”. In such circumstances, we sincerely do not think that such labeling could have altered their self-perception.

To go further, the literature developed on social labeling theory posits that for the self-concept to be altered among adults, the label has to be perceived as plausible. A label is not effective if it is perceived as contradictory to what people think about themselves (Tybout & Yalch, 1980). To avoid such perceptions, experimenters rely on people's actual behaviours or statements about behaviours to build a plausible label. As an illustration, Van der Werff, Steg, and Keizer (2014) justified an eco-friendly (resp. eco-unfriendly) label by using statements related to the 8 most- (resp. 8 least) performed environmental behaviours identified in a pre-test. Interestingly, behaviours or statements about previous behaviours that help justify the label plausibility do not need to be motivated by the trait stressed in the label. In Cornelissen et al. (2007) study, although participants may have selected the most ecological television set for its quality and price, stressing the prosocial dimension of this choice seemed to be sufficient for them to reconsider their original motivations due, at least in part, to pro-environmental dispositions. In addition, and in line with previous research conducted in a more general persuasion context (Baron, Baron, & Miller, 1973; Kumkale & Albarracín, 2004), Cornelissen et al. (2007) showed that social labelling needs a distraction task to be effective. When their protocol did not include any cognitive load, social labelling activated persuasion knowledge (Friestad & Wright, 1994), which led the individuals to reject the label (Burger, 1999).

Baron, R. S., Baron, P. H., & Miller, N. (1973). The relation between distraction and persuasion. Psychological Bulletin, 80, 310–323.

Burger, J. M. (1999). The foot-in-the-door compliance procedure: Amultiple-process analysis and review. Personality and Social Psychology Review, 3, 303–325.

Cornelissen, G., Dewitte, S., Warlop, L., & Yzerbyt, V. (2007). Whatever people say I am, that's what I am: Social labeling as a social marketing tool. International Journal of Research in Marketing, 24, 278–288. 

Friestad, M., & Wright, P. (1994). The persuasion knowledge Model: How people cope with persuasion attempts. Journal of Consumer Research, 21, 1–31. 

Kumkale, G. T., & Albarracín, D. (2004). The sleeper effect in persuasion: A meta-analytic review. Psychological Bulletin, 130, 143–172.

Tybout, A. M., & Yalch, R. F. (1980). The effect of experience: A matter of salience? Journal of Consumer Research, 6, 406–413.

Van der Werff, E., Steg, L., & Keizer, K. (2014). I am what I am, by looking past the present: The influence of biospheric values and past behavior on environmental selfidentity. Environment and Behavior, 46, 626–657. 

Now, we think you are perfectly right on one point. We did not provide enough evidence in the first version of our manuscript to support the following statement “the application of a “creative” label will be effective if the subjects do not perceive the persuasion attempt behind the label.” Therefore, we modified the following paragraph in the new version of our manuscript in order to be more convincing regarding the aforementioned statement:

In line with the literature, we contend that labeling an individual as “creative” could alter his or her self-perception, leading to a view of the self as creative, as soon as the label appears plausible. To build a plausible label, experimenters usually rely on people’s previous statements (Strenta & DeJong, 1981; Tybout & Yalch, 1980; Van der Werff, Steg & Keizer, 2014) or recent behavioral evidence (Cornelissen et al., 2007; Summers, Smith & Walker Reczek, 2016). When the label does not appear as plausible, persuasion knowledge (Friestad & Wright, 1994) is activated and the label is rejected (Burger, 1999; Cornelissen et al., 2007). As a conclusion, the persuasive intent behind the label must be masked for the self-perception to be altered among adults.

It is possible that “a “creative” label enhances self-perception of creativity while a “not creative” label erodes self-perception. The moderation analysis highlights this possibility. However, to have evidence for this statement, it is important to incorporate the measures of self-perception before and after the manipulation to demonstrate that with a positive label the self-perception goes up, while with “negative label” self-perception goes down.

We incorporated this in Study 3 (see above for the details):

The aim of Study 3 is to examine whether self-perceived creativity mediates the influence of a “not creative” label on the involvement in the creative task. To do so, we used Hayes' (2012) PROCESS macro (model 4) and 1000 bootstrapped samples but did not find any significant mediation between the construct. However, and as expected, the analysis shows that the application of the “not creative” label (vs. no label) both significantly damages post-label self-perceived creativity (F=12.19, p<.01, ηp2=.057, ω=0.97) and increases the time spent on the task (F=3.03, p<.05, ηp2=.015, ω=0.54).

Study 1. Samples size in study one is 76 people which suggests that the study is probably underpowered. Did you do power calculation and a sample size calculation?

We thank you for this comment. Originally educated in psychology, we more often use the criterium of the effect size to evaluate whether results are important enough. However, we do acknowledge that the power of the test appears today as a more robust criterium. 

To better satisfy the power criterium, we collected more data to complete Study 1 sample (please note that we now present Study 1 as Study 2). Precisely, we asked our market research institute to propose our study to 30 more panelists. These panelists were exposed to the “creative” label condition, which was previously underrepresented (n=28 versus n=48 for the control condition). Out of these 30 more respondents, 26 checked the manipulation regarding the label and were included in the database we provide along with our new submission (these new respondents are identified as 101, 102… 126 in the database). Following, the section dedicated to the presentation of the results of the study has been fully rewritten in our new submission to consider our data collection extension. Precisely, it shows that our results stay the same but that the addition of respondents allowed an increase in the power of our test, which reaches now 84,5% as now explicitly appears in the manuscript:

To do so, we used Hayes' (2012) PROCESS macro (model 4) and 1000 bootstrapped samples. The analysis first shows that the application of the label significantly influences post-label self-perceived creativity (�=.65, p<.01, ηp2=.067, ω=0.85).

Study 1. There was randomization in this study, why age, gender, education level, and self-perceived creativity are added as controls? There was no difference in these measures between groups, therefore these controls should be removed from the analysis

The control variables have been removed from the analysis. Most open, this has improved our results.

Study 1. The goal of the study was to show that the label influences participants’ self-perception of creativity, which was achieved. Why there was an analysis of self-efficacy? If it highlights the mechanism of how labels influence performance, why there is no measure of self-efficacy in Study 2? I would not state in the discussion that this research contributes to self-efficacy literature as there is no association between self-efficacy and the study conditions or performance.

We made sure not to speak about self-efficacy in Study 1 (previous Study 2) and checked that we were not stating that we were contributing to the literature on self-efficacy. Besides, as the addition of Study 3 minimize the role of creative self-efficacy in our whole paper, we also corrected the first paragraph of our introduction not to put too much focus on it. 

Study 1. Neither design of this study nor analysis speaks to this hypothesis. While this study could be a part of the research package, it does not directly speak to any of the stated hypothesizes. The measure of self-perception demonstrates rather that the manipulation worked as researchers planned it.

In the new version of our manuscript, we reversed Study 1 and Study 2. Study 1 aims at explicitly testing our conceptual framework. Study 2 and Study 3, introduced following your very relevant comment, aim at exploring further the psychological mechanisms at stake behind our results. This is how it now appears:

We will test our conceptual model in Study 1, before further exploring the psychological mechanism at play when labeling an individual as “creative” in Study 2, i.e. the activation of creative self-efficacy, or when labeling an individual as “not creative” in Study 3, i.e. an enhanced involvement in the creative task.

[…]

Study 2 and Study 3 extend these first findings on the impact of social labeling in creativity management by investigating the psychological mechanisms at play when applicating a “creative” label (Study 2) or a “not creative” label (Study 3).

Study 2. Where the participants' removal in this study? Did you have the same manipulation checks as in the previous one? Did participants remember the labels correctly?

We discarded participants who did not recall the manipulation using the same manipulation checks in our 3 experimental studies. We made sure to make this clear in the new version of the manuscript.

Study 2. Covariates should not be included in the experimental study, especially that there were no differences in these variables between conditions.

The control variables have been removed from the analysis. Most open, this has improved our results.

Study 2. Please report degrees of freedom in moderation study, it seems underpowered, especially for the continuous measure.

We calculated power for the moderation effect as follows:

To make this interaction effect easier to communicate (Fitzsimons, 2008), we dichotomized participants between higher and lower scorers on pre-label self-perceived creativity, using a median-split. We then carried out a classic ANOVA considering the manipulation (i.e. a “not creative” label vs. a “creative” label) and the dichotomized version of pre-label self-perceived creativity as between-subjects factors. This ANOVA replicated the significant interaction effect previously identified by the floodlight analysis between the positive/negative nature of the label and participants’ self-perceived creativity (F(1,96)=4.898, p<.05, ηp2=.049, ω=0.71). Planned contrast tests also revealed that the “creative” label enhanced individual creativity performance among participants scoring low on self-perceived creativity, who outperformed the “not creative” label group (6.18 vs. 5.45, F(1,49)=6.417, p<.01, ηp2=.116, ω=0.80). However, among participants scoring high on self-perceived creativity, the “not creative” group outperformed the “creative” label group, though in a marginally significant way (5.90 vs. 5.41, F(1,41)=2.387, p<.10, ηp2=.055, ω=0.45). Corroborating H3a but only marginally H3b, these results are depicted in Figure 4.

We acknowledge that the power is only medium regarding the influence of the “not creative” label. As we found that this symmetrical influence was clearly of interest, we chose to keep it in the new version of our manuscript and acknowledge this as a limitation in our final discussion.

This research contributes to the literature on social labeling. In showing the positive effect of a positive “creative” label on creative behaviors, it replicates previous findings observed in psychology and social marketing in an innovation management setting, thus extending the boundaries of social labeling’s relevance to the organizational field. In demonstrating the moderating effect of self-perceived creativity, it also offers a theoretical validation of social labeling theory itself, whose basic mechanism involves an alteration of individuals’ self-perception. Furthermore, in showing the positive effect of a negative “not creative” label, it corroborates the congruence hypothesis formulated by DeJong (1979) and Guéguen (2001), according to which a negative label only generates reactance when the label and the requested behavior are unconnected, such that the requested behavior offers no relevant opportunity to contradict the negative label. Additionnaly, Study 2 and Study 3 provide convincing empirical evidence about the mechanism at play behind the effect of each type of label. Still, we acknowledge here that the positive effect of a negative “not creative” label on individual creativity performance only appears marginal in Study 1 and therefore calls for replication.

However, would you prefer not to keep the focus on the “not creative” label in the manuscript, we would obviously follow your advice.

Study 2. While moderation analysis demonstrates the possibility of the highlighted in the introduction relationships, the further mediation analysis is needed to provide further evidence (see my comments above)

It appeared quite difficult to test the whole mediation in the same experiment. This is why we now provide 3 experiments. The first one demonstrates the effects on individual creativity performance and the two follow-ups investigate deeper the mechanisms at play. We hope that the addition of a new experiment to better explain the effects of labeling will convince you.

Study 2. I don’t see the value in following Hayes process with ANCOVA, you could remove it from the study and visualize your observations directly from the Process Model 1 procedure.

We do agree with the fact that such an analysis is not statistically necessary. Still we find it to be quite relevant to ensure a good communication of our experimental results. This is how we often proceed in our disciplinary field. Would you insist on getting rid of it, we naturally will follow your advice.

While I can see that the work of Bem, 1972 might be related to social labeling, It would be great if authors could elaborate on the connection between self-perception theory and social labeling.

To make it clearer, we do not state anymore that social labeling mechanism relies on Bem’s (1972) self-perception theory, which could be deceptive as social labeling does not require any previous behaviors to modify self-perception. We now insist on the re-attribution of dispositional properties that has been first developed by Bem (1972) and which allows to enlighten how social labeling actually works. This is how it appears in the new version of our manuscript:

Social labeling is “a persuasion technique that consists in providing a person with a statement about his or her personality or values (i.e. social label) in an attempt to provoke behavior that is consistent with the label” (Cornelissen, Dewitte, Warlop & Yzerbyt, 2007, p.279). Bem’s (1972) re-attribution mechanism explains the persuasion at stake in social labeling. Although self-observation of an individual’s own behavior provides the clearest information to make self-attribution of dispositional properties (Ouellette & Wood, 1998), labels provided by others can also be informative about a person’s traits and values and lead to their re-attribution to the self (Strenta & DeJong, 1981).

---

## [Decision Letter · Decision Letter 1]

16 Oct 2019

PONE-D-19-16350R1

Nudging individuals’ creativity using social labeling

PLOS ONE

Dear Dr Agogué,

While one of the reviewers is happy with your revision, the other one still has some major comments. Moreover, I have noticed that you did not address my comments, which I copy and paste below from my previous decision letter:

"I have now collected two reviews from two experts in the field. Both reviewers recommend major revisions. Therefore, I would like to invite you to revise the paper according to the reviewers' suggestions. Needless to say that all comments must be addressed. Particular attention, however, should be given to the issue raised by both reviewers regarding the smallness of the sample (especially in light of the current replicability crisis) and to the issue raised by Reviewer 1 regarding deception (please explain exactly which parts of the experiments involved deception and why you think this is not a problem for your experiments). Moreover, I would like to mention that social labels have been recently used also to impact people's decisions in economic games (https://www.sciencedirect.com/science/article/pii/S0022103116302098;
https://www.sciencedirect.com/science/article/pii/S0022103118302841;
http://journal.sjdm.org/17/171107/jdm171107.pdf;
http://journal.sjdm.org/19/190107/jdm190107.pdf). Note that I am the author of some of these papers. My decision on this manuscript will obviously be independent of whether you will decide to include or not these papers in your reference list. I have just thought that you might find them relevant.

Looking forward for the revision."

We would appreciate receiving your revised manuscript by Nov 30 2019 11:59PM. To enhance the reproducibility of your results, we recommend that if applicable you deposit your laboratory protocols in protocols.io, where a protocol can be assigned its own identifier (DOI) such that it can be cited independently in the future. For instructions see: http://journals.plos.org/plosone/s/submission-guidelines#loc-laboratory-protocols

We look forward to receiving your revised manuscript.

Kind regards,

Valerio Capraro

Academic Editor

PLOS ONE

Additional Editor Comments (if provided):

While one of the reviewers is happy with your revision, the other one still has some major comments. Moreover, I have noticed that you did not address my comments, which I copy and paste below from my previous decision letter:

"I have now collected two reviews from two experts in the field. Both reviewers recommend major revisions. Therefore, I would like to invite you to revise the paper according to the reviewers' suggestions. Needless to say that all comments must be addressed. Particular attention, however, should be given to the issue raised by both reviewers regarding the smallness of the sample (especially in light of the current replicability crisis) and to the issue raised by Reviewer 1 regarding deception (please explain exactly which parts of the experiments involved deception and why you think this is not a problem for your experiments). Moreover, I would like to mention that social labels have been recently used also to impact people's decisions in economic games (https://www.sciencedirect.com/science/article/pii/S0022103116302098;
https://www.sciencedirect.com/science/article/pii/S0022103118302841;
http://journal.sjdm.org/17/171107/jdm171107.pdf;
http://journal.sjdm.org/19/190107/jdm190107.pdf). Note that I am the author of some of these papers. My decision on this manuscript will obviously be independent of whether you will decide to include or not these papers in your reference list. I have just thought that you might find them relevant.

Looking forward for the revision."

Reviewers' comments:

Reviewer's Responses to Questions

**Comments to the Author**

1. If the authors have adequately addressed your comments raised in a previous round of review and you feel that this manuscript is now acceptable for publication, you may indicate that here to bypass the “Comments to the Author” section, enter your conflict of interest statement in the “Confidential to Editor” section, and submit your "Accept" recommendation.

Reviewer #1: All comments have been addressed

Reviewer #2: (No Response)

2. Is the manuscript technically sound, and do the data support the conclusions?

Reviewer #1: Yes

Reviewer #2: Yes

3. Has the statistical analysis been performed appropriately and rigorously? 

Reviewer #1: Yes

Reviewer #2: No

4. Have the authors made all data underlying the findings in their manuscript fully available?

Reviewer #1: Yes

Reviewer #2: (No Response)

5. Is the manuscript presented in an intelligible fashion and written in standard English?

Reviewer #1: Yes

Reviewer #2: Yes

6. Review Comments to the Author

Reviewer #1: (No Response)

Reviewer #2: Thank you for reviewing the manuscript. As before, I applaud your idea and interesting findings specifically with regard to the “non-creative” label. While I enjoy reading the updated version that indeed clearly communicated the framework. There are still changes that need to be made specifically with regard to the analysis of studies 2 and 3.

Abstract

Minor: Study 2 & 3 need to be better described in the abstract. Second to the last sentence is not clear.

Study 1: Please report the sample and the conditions that you included in the moderation analysis. Please remove analysis with dichotomization on page 16, but report the figure directly from PROCESS Model 1 with a continuous variable. To do so, you need to add in the PROCESS the option: “generate data for the plot” (the same relevant for Study 3). Please report conditional effects in Model 1, rather than re-running the analysis in ANOVA

Study 2: Please clarify whether you measured creative self-efficacy, before and after exposing people to labels. Did you control for pre-labeled self-perceived creativity in the mediation analysis?

Could you please provide a table that summarizes correlations between the variables in this study. The table will help to illustrate whether there is a collinearity problem between measures. The table should include pre-labeled self-perceived creativity, post-labeled self-perceived creativity, pre-(?) and post-self-efficacy, and study conditions.

I recommend reporting this study with repeated measure design, in which you have pre and post-self-perceived creativity with conditions as a factor. (See my comment below)

Study 3: The analysis of study 3 is confusing. It needs a picture with a “classic” triangle with a mediator on top. The conclusion as it stands now “both significantly damages post-label

self-perceived creativity (F=12.19, p<.01, ηp2=.057, ω=0.97)” looks like a result of ANOVA analysis but not Hayes Model 4. Please re-run the analysis as I suggested below.

Study 2 & 3 analysis and results are problematic as they stand now. Please state at the beginning that you are planning to test in two different studies conditions with creative and non-creative labels versus control conditions. Both studies need to have identical analyses answering three questions.

Question 1: whether a label changes self-perceived creativity before and after the manipulation. This could be accomplished with a repeated measure design and the conditions as a factor.

Question 2: whether low base self-perceived creativity moderates the effect of conditions on behavior (or its proxy, self-efficacy). In both studies, Hayes model 1 needs to be applied to answer this question. In model 1, conditions need to be included as independent variables, while base self-perception as moderator and dependent variables should be: self-efficacy (a proxy of behavior) in study 2 and a brick-task performance in Study 3. This analysis needs figures reported from Model 1.

Question 3: whether the change in self-perceived creativity influences participants’ behavior (or a proxy of it). A mediation analysis should be used. For study 2: conditions (independent variable)->post-self-perception controlling for pre-self perception (mediator) -> self-efficacy (dependent variable). For study 3: conditions (independent variable)->post-self perception controlling for pre-self perception (mediator) -> brick-task performance (dependent variable). This analysis needs figures.

Minor comment: please keep boosting samples consistent throughout the manuscript; report statistics with degrees of freedom e.g. t (df?)=1.98, p<.05; and confidence intervals.

With regard to the comments:

In study 1, you suggested “to make our label credible, participants were first asked to complete an initial questionnaire about their self-assessed ability to generate creative ideas” is it the same as “pre-label self-perceived creativity measure” in studies 2&3? If no, how did you mask a persuasive intend in study 2&3?

About the last comment: please do not dichotomize variables but report the statistics and figures from Hayes directly.

7. PLOS authors have the option to publish the peer review history of their article (what does this mean?). If published, this will include your full peer review and any attached files.

Reviewer #1: Yes: Harry M. Kaiser

Reviewer #2: No

---

## [Author Response · Author response to Decision Letter 1]

11 Jan 2020

Please see attached document for response to specific reviewer and editor comments.

---

## [Decision Letter · Decision Letter 2]

21 Jan 2020

PONE-D-19-16350R2

Nudging individuals’ creativity using social labeling

PLOS ONE

Dear Dr Agogué,

Thank you for submitting your manuscript to PLOS ONE. After careful consideration, we feel that it has merit but does not fully meet PLOS ONE’s publication criteria as it currently stands. Therefore, we invite you to submit a revised version of the manuscript that addresses the points raised during the review process.

Please find below the reviewer's comments. 

We would appreciate receiving your revised manuscript by Mar 06 2020 11:59PM. To enhance the reproducibility of your results, we recommend that if applicable you deposit your laboratory protocols in protocols.io, where a protocol can be assigned its own identifier (DOI) such that it can be cited independently in the future. For instructions see: http://journals.plos.org/plosone/s/submission-guidelines#loc-laboratory-protocols

We look forward to receiving your revised manuscript.

Kind regards,

Valerio Capraro

Academic Editor

PLOS ONE

Additional Editor Comments (if provided):

One of the reviewers suggests minor revisions before publication. Please address these remaining comments. I am looking forward for the final version.

Reviewers' comments:

Reviewer's Responses to Questions

**Comments to the Author**

1. If the authors have adequately addressed your comments raised in a previous round of review and you feel that this manuscript is now acceptable for publication, you may indicate that here to bypass the “Comments to the Author” section, enter your conflict of interest statement in the “Confidential to Editor” section, and submit your "Accept" recommendation.

Reviewer #2: All comments have been addressed

2. Is the manuscript technically sound, and do the data support the conclusions?

Reviewer #2: Yes

3. Has the statistical analysis been performed appropriately and rigorously? 

Reviewer #2: Yes

4. Have the authors made all data underlying the findings in their manuscript fully available?

Reviewer #2: (No Response)

5. Is the manuscript presented in an intelligible fashion and written in standard English?

Reviewer #2: Yes

6. Review Comments to the Author

Reviewer #2: The data are well described. I can easily follow the experiments and what was done for the experiment. Thank you for the changes.

Minor comments:

I assume that in Figure 3, error bars show the standard deviation. The figure should show a standard error, it allows readers to estimate the statistical difference from the figure.

Confidence interval 90 is unusual, the conventional way to report a confidence interval of 95 as it corresponds to conventional expectations of p < .05.

It is concerning that when you add pre-label self-perceived creativity in study 3 the indirect effect disappears. However, it could be an issue of low power. Can you please comment on the effect sizes, e.g. what is the mean difference between conditions in post-label self-creativity? and what is the mean difference between conditions in individual creativity performance? That information will allow a reader to estimate the magnitude of the effect of the "non-creative" label.

7. PLOS authors have the option to publish the peer review history of their article (what does this mean?). If published, this will include your full peer review and any attached files.

Reviewer #2: No

---

## [Editor Report · Decision Letter 3]

28 Jan 2020

Nudging individuals’ creativity using social labeling

PONE-D-19-16350R3

Dear Dr. Agogué,

We are pleased to inform you that your manuscript has been judged scientifically suitable for publication and will be formally accepted for publication once it complies with all outstanding technical requirements.

With kind regards,

Valerio Capraro

Academic Editor

PLOS ONE
---

## [Editor Report · Acceptance letter]

6 Feb 2020

PONE-D-19-16350R3 

Nudging individuals’ creativity using social labeling 

Dear Dr. Agogué:

I am pleased to inform you that your manuscript has been deemed suitable for publication in PLOS ONE. Congratulations! Your manuscript is now with our production department. 

With kind regards,

on behalf of

Dr. Valerio Capraro 

Academic Editor

PLOS ONE